



# Uncertainties in the atmospheric loading to ice-sheet deposition for volcanic aerosols and implications for forcing reconstruction

Ya Gao[1,2], Chaochao Gao[1,2]

[1]College of Environmental and Resource Science, Zhejiang University, Hangzhou 310058, China
[2]Zhejiang Provincial Key Laboratory of Organic Pollution Process and Control, Hangzhou 310058, China

*Correspondence to*: Chaochao Gao, gaocc@zju.edu.cn

**Abstract.** Volcanic radiative forcing reconstruction is an important part of paleoclimate simulation and attribution efforts, and the conversion factor used to transfer ice core-based sulfate observation into stratospheric volcanic aerosol loading (hereafter referred to as atmospheric-loading to icecap-deposition, *LTD*) is critical for such reconstruction. This study revisits
the Pinatubo-based *LTD* applied in the CMIP5 and CMIP6 volcanic forcing, by using 54 polar ice core records of Tambora deposition and a Monte Carlo sampling model. A set of Tambora-based *LTD* with associated uncertainties are obtained, which corrects the bias of over-representing the west Antarctic. New *LTD*s for Pinatubo and Agung are calculated using 18 and 24 Antarctic ice core observations, respectively, and the uncertainties are evaluated against the Monte Carlo characterization with varying ice core numbers. The comparison of Southern Hemispheric *LTD* among Tambora, Pinatubo
and Agung suggests that, the conversion factor may vary significantly among different eruptions. Even larger uncertainty is revealed when compare the ice-core-based conversion factor with the model results. Both results suggest systematic and stochastic causes that are difficult to anticipate, and call for precaution when single conversion factor is used for reconstruction.

## 1 Introduction

Stratospheric volcanic eruption is an important cause of natural climate variability, and polar ice preserves the nature, including timing and magnitude of the historical eruptions long before human observation. For the ice-core volcanic achieves to be utilized in climate models, however, one has to convert the amount of deposited volcanic sulfate in the ice caps back to the stratospheric sulfate mass loading. This inverse reconstruction may introduce substantial uncertainties, due to the discrepancy in ice core volcanic deposition measurements and perhaps more importantly, the limitations in the
conversion factor to transfer the ice core observation into the stratospheric volcanic sulfate loading (hereafter referred to as the *LTD* factor).

Previous studies have tried to derive the *LTD* factor combing different lines of observation and model simulations. Clausen and Hammer (1988) pioneered the use of bomb test debris to calculate the *LTD* factor for Greenland ice cores, based on the assumption that the transport and deposition of volcanic aerosols are analog to those of bomb test debris on a large scale.



Cole-Dai and Mosley-Thompson (1999) utilized the ratio between 1991 Pinatubo volcanic depositions in six South Pole ice
cores and its atmospheric aerosol loading to linearly convert the ice core volcanic signals to their stratospheric loadings,
assuming similar transport and deposition pattern for all tropical eruptions. Crowley (2000) and Ammann et al. (2003)
applied the same approach but different reference events of 1883 Krakatau and 1815Tambora, respectively, as the empirical
scaling. Crowley (2000) applied an additional 2/3 power dampening factor, to account for the self-limiting effects of sulfate

aerosols for the eruptions larger than Pinatubo (Pinto et al., 1989).

Using the satellite observations of the atmospheric Pinatubo aerosol loading and 19 ice core depositions records in Antarctic,
combined with the bomb-test derived factors for Greenland, Gao et al. (2007) obtained the conversion factor for tropical
eruptions assuming all tropical eruptions following the same atmospheric transport pattern of 1991 Pinatubo volcanic aerosol.
The results were verified in the GISS Model E simulation (Gao et al., 2007) and the conversion factors have been utilized in

the ice-core-based volcanic forcing reconstruction of Gao et al. (2008) and Sigl et al. (2015), which has been widely used in
the CMIP5 and CMIP6 model simulations, respectively.

In recent years, the coupled chemistry-climate models have been utilized to estimate the conversion factors. Toohey et al.
(2013) tested the assumption of directly proportional relationship between the stratospheric volcanic aerosol loading and ice
sheet deposition using the MAECHAM5-HAM aerosol-climate model, and the results show excellent spatial correlation but

4-5 times larger deposition fluxes compared with ice core observations. Marshall et al. (2018) further calculated the
atmospheric burden to ice core deposition conversion factors using MAECHAM5- HAM and three additional coupled
chemistry-climate models, and found the conversion factors to differ by a factor of five for Northern Hemisphere and by an
even larger factor of 15 for Southern Hemisphere among the models. The results suggest the difficulty in giving the accurate
estimation of the conversion factors for volcanic clouds with the current aerosol-climate models.

Due to the limited number (0 in Greenland and 6 in Antarctic) of ice core sulfate observations for the late 20[th] century, the
conversion factor for Pinatubo is derived combining three methods, its uncertainties and applicability across different
eruptions have been largely overlooked. Recent improvements to the ice core record, especially the largely static and high-
depth-resolution volcanic sulfate records from East Antarctic (Sigl et al., 2014), and the coupled chemistry-climate modeling
studies of Tambora or Tambora-size eruptions, make the revisit of the conversion factor and evaluation of its uncertainties

possible.

In this study, we aim to first derive a new set of *LTD*s for Tambora eruption based on 22 Greenland and 32 Antarctic - the
most comprehensive collection of ice core sulfate observations; and then apply a Monte Carlo random sampling model to
characterize the uncertainty in *LTD*s associated with the varying number of ice core observations. The latter is then utilized
to estimate the uncertainty range of *LTD*s for Pinatubo and Agung, which are based on 18 and 24 Antarctic ice-core-records,

respectively. The set of Tambora-based *LTD*s and uncertain estimates will be consistent for Antarctic and Greenland ice core
data in terms of methodology, and can be compared both with the multimodal-derived conversion factors (Marshall et al.,
2018) and other ice-core-derived factors in a more systematic framework.



## 2 Data and Methods

By measuring the amount of volcanic sulfate that was deposited in polar ice sheets, in theory one could an inverse
calculation of the LTD factor as the following Eq. (1):

$$LTD = L \div D \tag{1}$$

where $L$ is the stratospheric volcanic mass loading (usually in Tg of $SO_2$ or sulfate aerosols); $D$ is the icecap average of the
volcanic deposition measured in each ice core record (usually in kg $km^{2-}$ of none-sea-salt $SO_4^{2-}$).

### 2.1 Volcanic deposition in polar ice cores

Calculation of the volcanic deposition, if start with the raw ice core measurements, involves multiplication of ice or snow
concentration ($c$), ice or water equivalent density ($d$), and the deposition thickness ($h$) as shown in Eq. (2):

$$D = c \times d \times h \tag{2}$$

In this study we maintain the original ice core records used to derive the Pinatubo-based LTD, which in Greenland includes
six PARCA cores (Mosley-Thompson et al., 2003), 12 cores from Clausen and Hammer (1988), and 6 cores original to Gao
et al. (2007). Therefore, the volcanic depositions for selected events are obtained directly from these records. In addition, we
also include two newly obtained northern Greenland cores – NEEM2011S1 and Tunu2013 (Sigl et al., 2015) in the
calculation. For these two records, we apply the same volcanic signal extraction procedure as Gao et al. (2007), that is we
identify the volcanic signals by applying a high-pass loess filter remove the influence of background concentration and
extract the peaks that exceed twice the 31-year absolute running median. Table 1 lists the general information of these
Greenland ice cores, and Figure 1 shows the site map and Tambora volcanic deposition of these ice cores. The addition of
NEEM2011S1 and Tunu2013 significantly improved the sampling coverage of the low-accumulative Northern Greenland.
In Antarctic, we nearly double the number of ice-core records from the 17 in Gao et al. (2007) to 32 by including the
annually dated WDC06A core from West Antarctic and 14 additional AVS-2K cores mainly from East Antarctic (Sigl et al.,
2013 & 2015). Table 2 lists the information and Figure 2 shows the location and average accumulation rate of these
Antarctic ice cores. Same procedure is applied to the WDC06A and AVS-2K cores, so that all the ice core volcanic
depositions are obtained with the same criteria.

### 2.2 Calculation of the LTD factor

The 1815 Tambora eruption in Indonesia is chosen as the representative event to calculate the LTD factor, because its
characteristics are relatively well known and the signals are detected in almost all ice core records. We also calculate the
LTD factor of the 1963 Agung and 1991 Pinatubo eruptions, because they are the two recent events with comparable
magnitudes and the latitudinal location of Agung is very close to Tambora (Figure 3).
We calculate the ice sheet mean deposition by taking the simple average of all records without consideration of their
geological location and distribution, and in Greenland the simple-average Tambora deposition flux is 57 kg km$^{-2}$. For





Antarctic, we take one step further by multiplying the average deposition of West Antarctic Peninsula and East Antarctic
with 0.2 and 0.8, respectively, to reflect the size difference of these two areas (Sigl et al., 2015). The LTD factor for the
selected three events are obtained following Eq. (1).

### 2.3 Monte Carlo random sampling of the ice core records and uncertainty estimation of the LTD factors

We retrieve the volcanic sulfate signals in 54 records, including 22 from Greenland and 32 from Antarctic ice cores, which
we believe is the most comprehensive ice core - based observation (Table 1 & 1). Nevertheless, this coverage of ice-core-
observation is available only for limited eruptions such as Tambora and Laki, whereas the Tambora volcanic deposition flux
varies as large as one order of magnitude among cores in both Greenland and Antarctic (Figure 1 & 2). It is therefore crucial
to assess the representativeness of ice core observations, in terms of both individual ice-core or ice cores from certain regions
and the threshold number of ice core observations. The results are then applied to evaluate the representative and the
uncertainties in the associated LTD factors.

We conduct Monte Carlo random sampling of the volcanic records with varying ice core numbers, for each selected event in
Antarctic and Greenland. This process is repeated 10,000 times to build a random distribution for the average Antarctic or
Greenland volcanic deposition, against which the associated LTD factor for individual event is calculated by dividing the
estimated magnitude of total stratospheric sulfate aerosol loading by the average icecap.

## 3 Results and Discussions

### 3.1 New LTD estimated with Tambora ice core records

The April 1815 eruption of Tambora (8.25° S, 118.00° E; Figure 3) released 60 - 80 Tg of $SO_2$ into the stratosphere, making
it one of the largest explosive eruptions in the Common Era (Self et al., 2004; Gertisser et al., 2012). It is also the most
widely studied eruption in terms of ice core observation, model simulation, proxy reconstruction, and climatic and
socioecological aftermaths assessment (Luterbacher and Pfister, 2015; Raibie et al., 2016; Gao et al., 2017; Brönnimann et
al., 2019).

Due to its explosive nature and large magnitude, Tambora volcanic deposition is widely observed in the polar ice sheets and
used as the first order reference layer for ice core dating (Figure1&2). Figure 3 shows the latitudinal and longitudinal
location, accumulation rate, and Tambora sulfate deposition of these two group of records, from which we can see that all of
the MS15 ice cores except WDC06A and WDC05Q are located in inland East Antarctic within 74-84° S latitude band. The
mean accumulation rate of the MS15 ice cores is 0.068 meter per year, only one third of that for GC07 cores (0.20 m/a). As a
result, the high-cumulative west Antarctic was over represented in the LTD calculation for Pinatubo in Gao et al. (2007),
which we will discuss in the next section.

The composites, after correcting for area-difference, show relatively similar sulfate deposition in Greenland (57 kg km$^{-2}$) and
Antarctic (47 kg km$^{-2}$). This suggests that the hemispheric partition of Tambora sulfate aerosols is probably symmetric.



Therefore, if we take low size estimation of Tambora eruption, i.e., 60 Tg $SO_2$ as the total amount of sulfate gases injected into the stratosphere and divide the values equally into each hemisphere. This results in 61Tg of sulfate aerosols (assuming 75wt. % $H_2SO_4$ in water) in each hemisphere, respectively. The obtained ratio between the stratospheric sulfate loading and the average amount of sulfate deposited on each ice sheet for Greenland (hereafter referred to as NH-LTD$_T$) is $1.07 \times 10^9$ km$^2$

and for Antarctic (hereafter referred to as SH-LTD$_T$) is $1.29 \times 10^9$ km$^2$.

### 3.2 Characteristic of LTD$_T$ due to different ice-core sampling

Toohey and Sigl (2017) suggested that the uncertainties in the conversion factor are composed of systematic uncertainties, static errors that potentially causing global bias in forcing estimation, and random differences that are specific to individual volcanic events or forcing reconstruction. The LTD$_T$ values are subject to systematic uncertainties, for example, uncertainty

in the estimation of stratospheric sulfate aerosol injection will directly influence the LTD$_T$ values and the associated forcing reconstructions. If we apply the larger value of existing estimation 80 Tg $SO_2$, then NH-LTD$_T$ and SH-LTD$_T$ becomes $1.65 \times 10^9$ km$^2$ and $1.72 \times 10^9$ km$^2$, respectively

The random uncertainties are commonly associated with the variation of volcanic aerosols dispersion from case to case, and the finite sampling of ice cores through time. We therefore repeat the LTD$_T$ calculations for finite sampling of ice cores, by

applying the Monte Carlo random sampling with various sample sizes (Table 1 & 1) to build a distribution range and associated probability (Figure 4). The results show that, the distributions of LTD$_T$ with different ice-core sample sizes are approximately normal, with slightly longer tails toward the right (Table 3). Therefore, we could use the mean ($\mu$) and standard deviation ($\sigma$) to characterize the LTD$_T$ values.

First of all, the distribution of differently-sampled LTD$_T$ largely overlap with each other. The NH-LTD$_T$ and SH- LTD$_T$

values derived from the full set of Greenland and Antarctic ice core observations also lie very close to the mean of the Monte Carlo simulated distributions (Figure 4). Both results suggest the robustness in the estimated mean value w.r.t the sampling of ice cores. Secondly, the standard deviation of LTD$_T$ decreases with 1/sqrt (N), suggesting reduction of the uncertainties as the number of ice core records increases (Figure 4).

The fluctuation of LTD$_T$ values within $\mu \pm \sigma$ also suggests that the noise is white and as long as $\sigma$ decreases with 1/sqrt (N)

any additional core will reduce the noise. Therefore, the precision of LTD values is related to the limit in the number of cores. For example, if we want to reach 65% of precision, 14 Greenland and 20 Antarctic ice core records will be necessary, respectively (Table 3).

### 3.3 SH-LTD of Pinatubo based on updated ice core records and Monte Carlo characterization

For comparison, we recalculate the conversion factor SH-LTD for the Pinatubo eruption (hereafter referred to as SH-LTD$_P$)

using 18 available ice core records (Table 2), and the resulting value is $1.87 \times 10^9$ km$^2$. Assuming that the Monte Carlo characterization of SH-LTD$_T$ with the change in ice core ensemble number (Table 3) is the same for all low latitude



eruptions, we could expand the possible range of the SH-LTD$_P$ value to be $(1.87\pm0.145) \times10^9$ km$^2$. According to Table 3, since the noise is white the precision of the estimated SH-LTD$_P$ value based on 18 cores is 56 %.

Therefore, the conversion factor for the Pinatubo eruption is larger than that of Tambora, assuming 60 Tg SO$_2$ injection (Figure 5). One possible reason for the difference is that, the Antarctic cores with the highest Tambora deposition, i.e., Siple Station, Dyer, and G15, do not have record for Pinatubo deposition, therefore probably reduces the Antarctic average Pinatubo deposition and increases the conversion factor. If we remove the three ice core records from the Tambora simulation, the SH-LTD$_T$ value would increase by 1/3, for example, from $(1.08\pm0.056)\times10^9$ km$^2$ to $(1.44\pm0.077) \times10^9$ km$^2$ assuming 60 Tg SO$_2$ injection. The variation in ice core depositions therefore introduce another layer of uncertainty. On the

other hand, the *SH-LTD$_P$* value largely overlaps with *SH-LTD$_T$* assuming 80 Tg SO$_2$ injection $((1.74\pm0.09) \times10^9$ km$^2$; Figure 5).

Using the satellite observations of the atmospheric Pinatubo aerosol loading and 19 ice core depositions records in Antarctic, Gao et al. (2007) also calculated the conversion factor for Antarctic ice core-derived volcanic signals (hereafter referred to as *L$_P$*, Table 4). The new conversion factor *SH-LTD$_P$* $((1.87\pm0.145) \times10^9$ km$^2$) is also larger than $L_P$ $( (1.0\pm0.25) \times10^9$ km$^2$). One

possible reason being that only six out of the 19 Antarctic records used in Gao et al. (2007) cover the vast regions of inner and east continent, and the accumulation rates in the west Antarctic cores are on average one order of magnitude larger than the east Antarctic cores. The uneven distribution of the ice core records, coupled with the large spatial variability of volcanic deposition (Zielinski et al., 1995; Traufetter et al., 2004; Gautier et al., 2016) may have biased the conversion factors. This is also the case between $L_P$ and the conversion factor obtained in Toohey and Sigl (2017) who repeated the calculation using an

updated set of ice core records (Sigl et al. 2015), and the result is 35 %-50 % larger than $L_P$.

**3.4 SH-LTD of Agung and comparison among the three eruptions**

Similar procedure is applied to obtain the *SH-LTD* for the 1963 Agung eruption, and the resulting conversion factor is $0.95\times10^9$ km$^2$ using 24 ice core records available (Table 2). According to Table 3, with 75 % precision the Monte Carlo simulation suggests the distribution range of *SH-LTD$_A$* to be $(0.95\pm0.048) \times10^9$ km$^2$.

Comparison among the three eruptions suggests the possibility that, the *LTD* factor may vary for individual eruption, depending on the volcano location, eruptive magnitude, etc. Agung volcano lies close to Tambora (Figure 6), while its eruption size is much smaller in terms of the SO$_2$ injection, therefore more sulfate aerosols may have stayed in the atmosphere longer and reached the ice sheets. Besides, observations indicate 2SH:1NH dispersion of the Agung aerosols. Although we have accounted for the hemispheric partitioning difference in calculating the stratospheric loading,

disproportionally more Agung debris could have reached Antarctic and dwarf the conversion factor. Pinatubo is about 20-degree north of Tambora, but observations show that the volcanic cloud disperse more or less evenly between the two hemispheres. It is difficult to track the hemispheric partitioning of Tambora cloud, although the ice core records tend to suggest even distribution. The dispersion and transport of individual volcanic cloud are therefore difficult to anticipate, and their potential influence on *LTD* is hard to quantify.






## 4 Comparison with the conversion factors obtained from other sources

### 4.1 Conversion factors obtained from the bomb test observations

A serial nuclear bomb tests were conducted from 1945 until 1980. In particular, the U.S. conducted two tests in the Pacific sites of Bikini (11° 35′ N 165° 23′ E, in 1952 CE) and Enewetak (11° 30′ N 162° 15′ E, in 1954 CE) during one of the most

active period 1954-1958 CE (Bennett, 2002), and the location of these two test sites are in close latitudinal approximation of the Pinatubo (Figure 6). The explosion columns of these bomb tests are 20 km high and characterized by near instantaneous release and global dispersal of the volatile mass. The stratospheric residence time of the radionuclides is about 1-3 years, resembling the stratospheric residence time of the volcanic aerosols (Bennett, 2002; Robock, 2000).

Radioactive debris from the bomb tests were transported to polar ice sheets and preserved in the ice, and can be identified by

total $\beta$ activity examinations. Clausen and Hammer (1988) measured both the Tambora volcanic deposition and the total $\beta$ activity of the 1952-54 tropical Pacific bomb tests in 16 Greenland ice core and obtained the first sets of bomb test-based conversion factor for Greenland (hereafter referred to as $L_{\beta-1982}$).

Gao et al. (2007) revisited the calculation with the updated United Nations Scientific Committee on the Effects of Atomic Radiation (UNSCEAR 2000) report, and used only the fission yields in stratosphere to mimic the volcanic sulfate loading

(hereafter referred to as $L_{\beta-2000}$, Table 4). The obtained conversion factor (hereafter referred to as $L_{\beta-2000}$, Table 4) differ from the original $L_{\beta-1982}$ by a factor of two for the low NH latitude bomb tests, because about half of the fission ends up in the stratosphere. Combining the bomb-test derived factors for Greenland and the Pinatubo-observation derived factors for Antarctic, Gao et al. (2007) obtained the conversion factor for tropical eruptions (hereafter referred to as $L_P$, Table 4).

### 4.2 Conversion factors obtained from model simulations

Toohey et al. (2013) calculated the deposition efficiency (i.e., the ratio between the hemispheric maximum stratospheric aerosol loading and the volcanic sulfate flux in Greenland or Antarctic), for the MAECHAM5-HAM aerosol-climate model. The model ensemble conversion factor for a Tambora - size eruption is $0.2 \times 10^9$ km$^2$ for both Greenland and Antarctic, regardless of the season.

Marshall et al. (2018) compared the Tambora volcanic sulfate deposition in four global aerosol-climate models, i.e.,

MAECHAM5- HAM, CESM1-WACCM, SOCOL-AER, and UM-UKCA, to that in ice core observations and calculated the burdening-to-deposition factors (hereafter referred to the original reference as *NH-BTD* and *SH-BTD*). These BTD values, together with that from Toohey et al. (2013) are listed in Table 4, and serve as reference results for the uncertainty evaluation for the ice-core-based *LTD* factors.



### 4.3 Uncertainties in the conversion factors obtained from different methods

For Southern Hemisphere, none of the four aerosol-climate model simulated conversion factors fall within the range of the newly obtained $SH\text{-}LTD_T$, despite the fact that the 4 model-mean $SH\text{-}BTD_T$ appear to be in good agreement (Figure 5, Table 4). $SH\text{-}BTD_T$ itself varies among the four models by up to a factor of 15 (Marshall et al., 2018). CESM-WACCM derived $SH\text{-}BTD_T$ is the closest to the ice-core derived $SH\text{-}LTDs$. In Northern Hemisphere, the UM-UKCA derived $NH\text{-}BTD_T$ fall within the range of $NH\text{-}LTD_T$. Both of them, nevertheless, have $BTD_T$ of the other hemisphere way off the $LTD_T$ ranges. The

conversion factor for Tambora and Pinatubo based on previous GISS simulations (Gao et al. 2007) are also much smaller than their $LTD$ counterparts (Figure 5), echoing the general model tendency to transport more flux toward polar regions.

Different from the symmetric partition assumption based on ice core observations, all four aerosol-climate models tend to keep more Tambora volcanic sulfate aerosols in Southern Hemisphere (Marshall et al., 2018), likely as a result of the 8.25° S latitudinal location of the volcano. None of the model simulated volcanic deposition in ice sheets is in close agreement with

ice core observation, therefore it is also difficult to match ice core observations with individual model to judge model performance or confirm model-proxy consistency in the conversion factors.

We also compare $NH\text{-}LTD_T$ with the Gao et al. (2007) estimation of $L_{\beta\text{-}2000}$. As we can see from the pink shading in Figure 5, the $NH\text{-}LTD_\beta$ values, either symmetric or 2NH:1SH partitioning, are smaller than $NH\text{-}LTD_T$. The $L_{\beta\text{-}}2000$ value under the 2NH:1SH partitioning assumption is, however, close to $SH\text{-}LTD_A$. Given that both values are based on the ice core

observations of the same hemisphere and assuming a two third hemispheric partitioning of the event debris, the closeness in results confirms that the $LTD$ factor may vary depending on eruption magnitudes and locations.

In summary, the conversion factors obtained among different lines of estimations contain large uncertainties. Some of the uncertainties are systematic, likely associated with the methodologies applied to derive the factor. For example, climate models tend to give small conversion factor due to the large poleward transport of stratospheric aerosols. Or without proper

area-weighting, the Antarctic-mean deposition maybe overestimated due to the disproportionally-dense ice core sampling in the high-accumulative West Antarctic Peninsula. Others are more specific and therefore difficult to evaluate. For example, the specific transport and deposition characteristic of individual eruptions, uncertainties in the volcanic deposition from core to core and from event to event, uncertainties also in ice core signal measurements.

### 5 Conclusions

The existing volcanic reconstructions commonly rely on the Pinatubo-based conversion factor to estimate the radiative forcing of historical volcanic eruptions. This study revisits the conversion factor, using a large collection of polar ice core records of Tambora deposition and a Monte Carlo sampling model. Uncertainties associated with the $LTD$ factor and its applicability to the volcanic forcing reconstruction are examined, by both across-methodology-comparison between the ice core-based estimation and multi-model simulations, and across-event-comparison among Tambora, Pinatubo, Agung

volcanic aerosols, and bomb test debris.

The newly obtained *LTD*s for Tambora is the first set of conversion factor that are based on consistent methodology and the most comprehensive collection of ice core observations. It also contains uncertain range estimated from Monte Carlo sampling, and corrects the bias of over-representing the west Antarctic in our previous Pinatubo-based estimation. Monte Carlo simulation suggests that different ensemble numbers of the ice core records introduce white noise in the *LTD*
estimation, and additional record tends to narrow the uncertainty range of *LTD*. For example, the standard deviation of *NH-LTD$_T$* and *SH-LTD$_T$* reduces from 32 % and 21 % to 10 %, respectively, when the sampled ice core number increases from 8 to 18 in Greenland and from 14 to 24 in Antarctic. We therefore calculate the *SH-LTD* for Pinatubo and Agung using the available 18 and 24 ice core records, and obtained the corresponding uncertainty ranges according to the Monte Carlo characterization.

The comparison of *SH-LTD* among Tambora, Agung, and Pinatubo suggests that, the conversion factor may vary significantly for individual eruption, depending on the volcano location, eruptive characteristic and magnitude, etc. The magnitude-induced variation is also simulated by the MAECHAM5-HAM model (Toohey et al., 2013), albeit only matters for eruptions larger than Tambora. It is therefore important to acknowledge that none *LTD* estimated from a single eruption could probably represent all the eruptions. On the other hand, until the aerosol-climate models could accurately and
consistently simulate the volcanic clouds, and establish a systematic characterization of the dispersion and deposition pattern of volcanic aerosols of different location, magnitude, and seasonality, etc., we will have to rely on the reference eruption(s) for forcing reconstruction. The newly obtained conversion factors could serve as additional choices in future volcanic forcing reconstruction work, especially when Tambora, Pinatubo, or Agung is utilized as a reference.

## Data Availability

The supplementary dataset and information are accessible at http://www.geodoi.ac.cn (Gao C C and Gao Y, 2020, DOI: 10.3974/geodb.2020.07.07.V1).

## Acknowledgments

The work is supported by the National Natural Science Foundation of China (41875092). We sincerely thank all the colleagues who drilled the ice cores and generated the original volcanic signal data, without whom this work and our
previous works on volcanic forcing reconstruction would not be possible. We also appreciate the anonymous reviewers for their constructive comments and suggestions which helped to improve the paper.

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

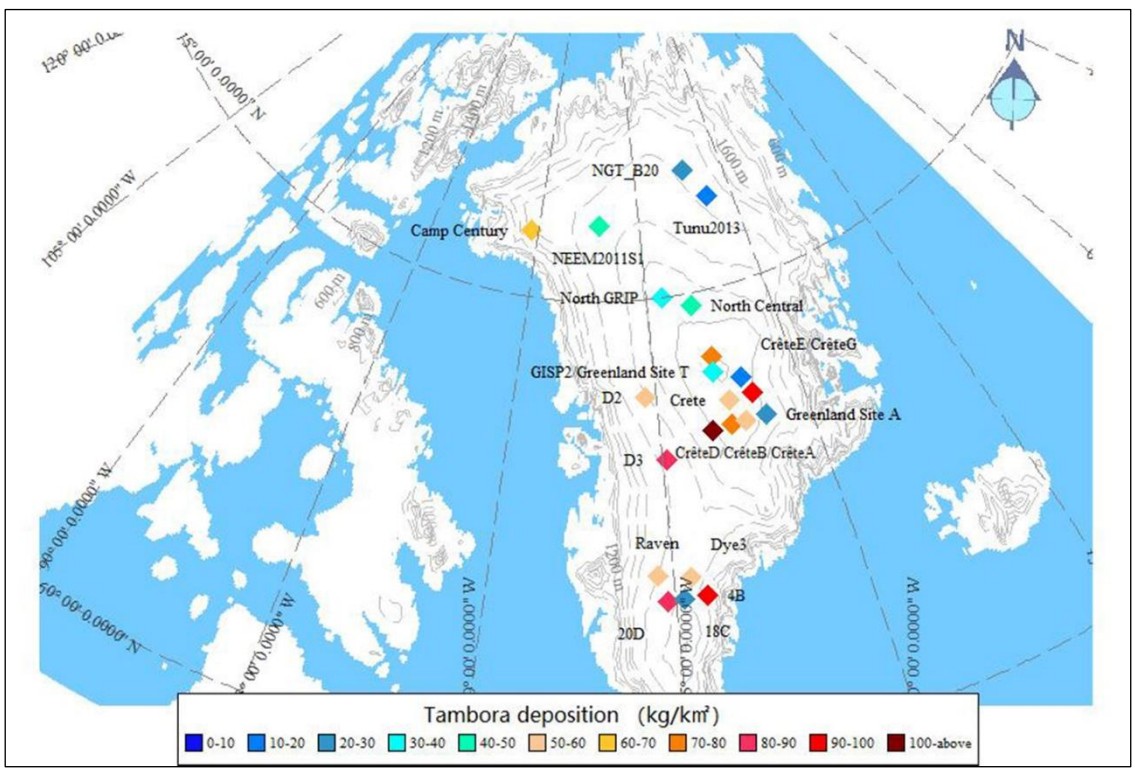

**Figure 1. Location and Tambora volcanic deposition of the Greenland ice core records listed in Table 1.** The figure was drawn by plotting the site markers on a base map obtained from © Google Maps 2021.





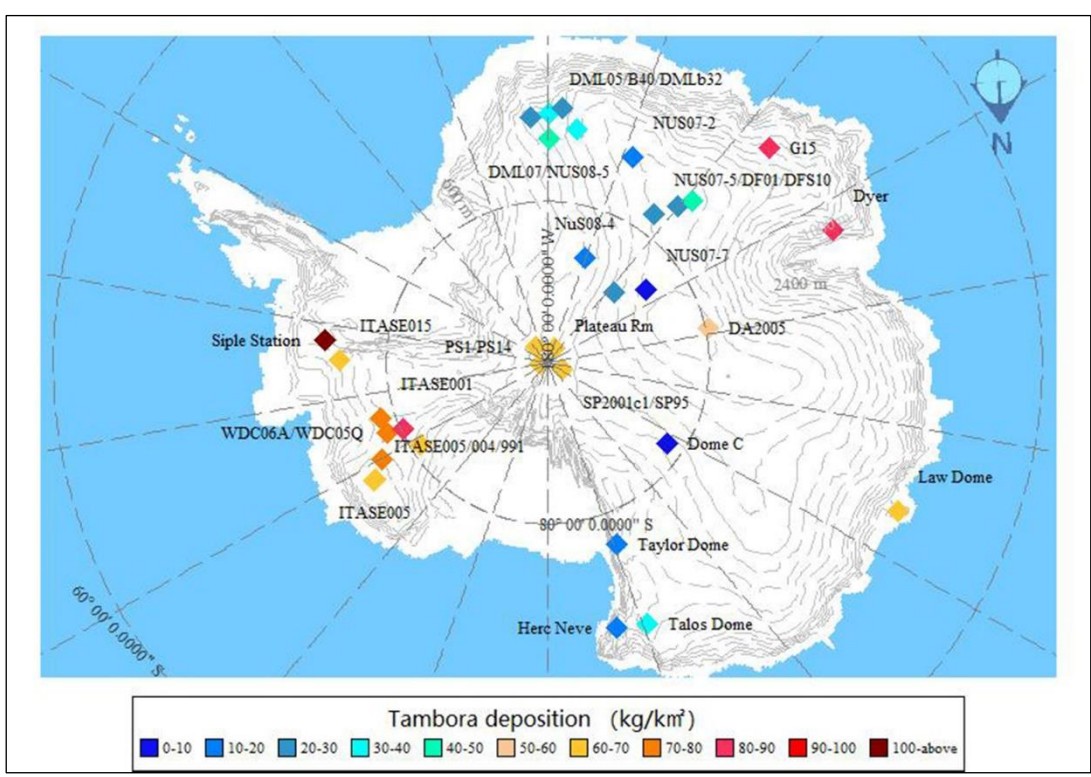

**Figure 2. Location and Tambora volcanic deposition of the Antarctic ice core records listed in Table 2.** The figure was

drawn by plotting the site markers on a base map obtained from © Google Maps 2021.





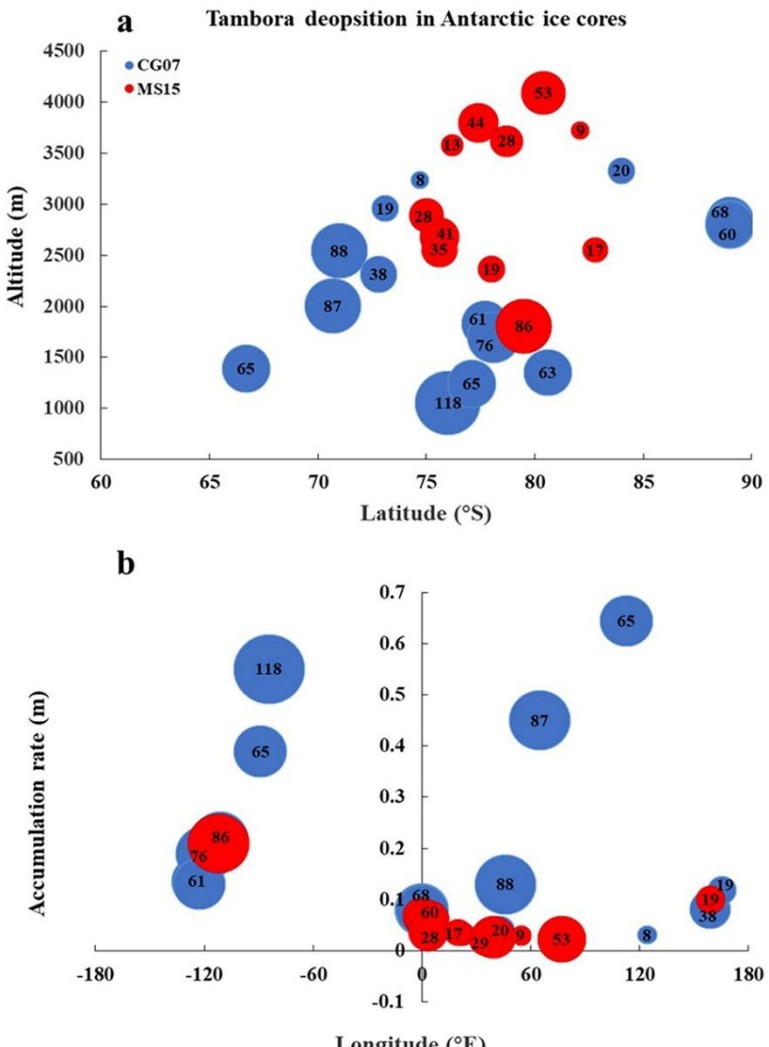

**Figure 3. Difference in the Tambora deposition between the Antarctic ice cores used in the Gao et al. (2007, referred to as CG07 and represented as the blue bubbles) and Sigl et al. (2015 referred to as MS15 and represented as the orange bubbles**) reconstruction. The size of the bubbles and values inside each bubble indicates the deposition (kg/km$^2$). Some ice core records overlap with the others therefore are not visible in the figure.





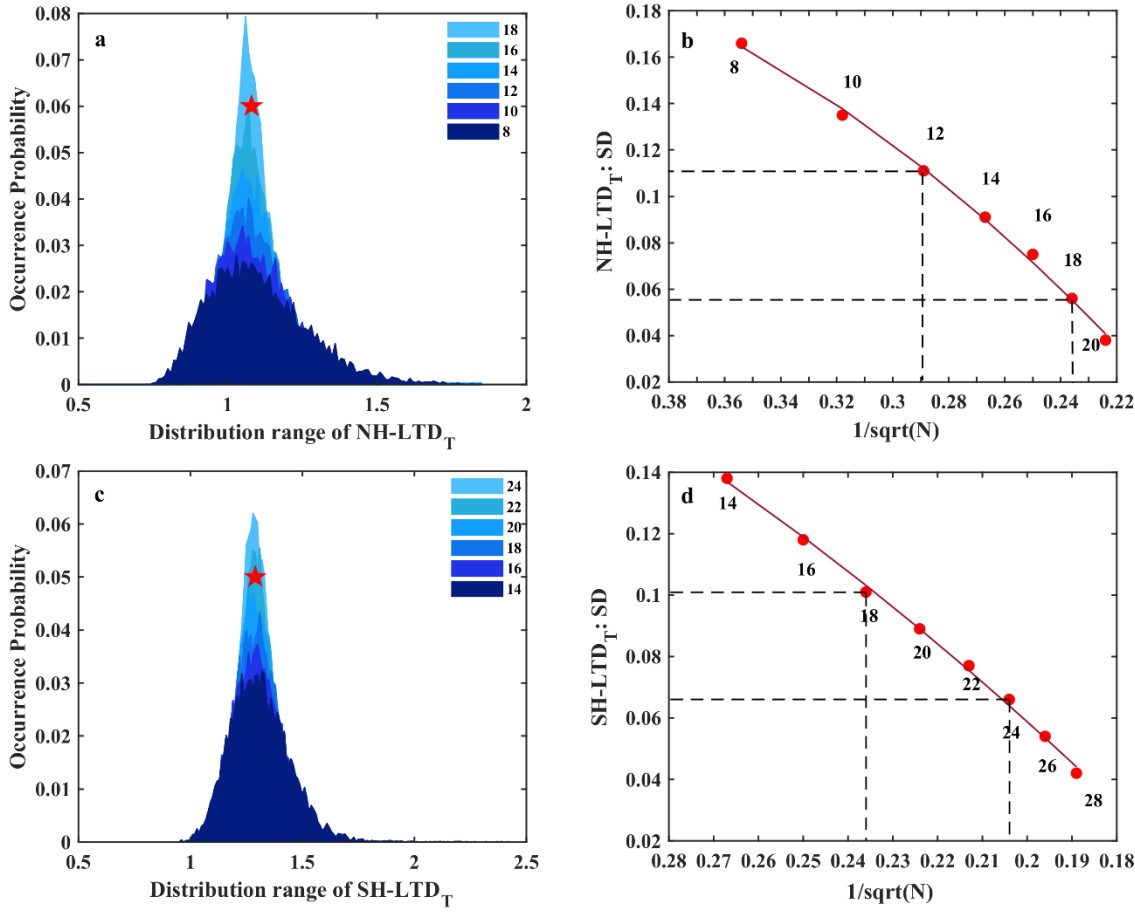


**Figure 4. (a, c) Distribution of NH-LTD$_T$ and SH-LTD$_T$ estimated by Monte Carlo random sampling of selected number of ice core records; and (b, d) the standard deviation of LTD decreases with 1/sqrt (N), where N is the number of cores.** The red star in panel a and c represents the NH-LTD$_T$ and SH-LTD$_T$ values estimated with the full set of available ice core records.





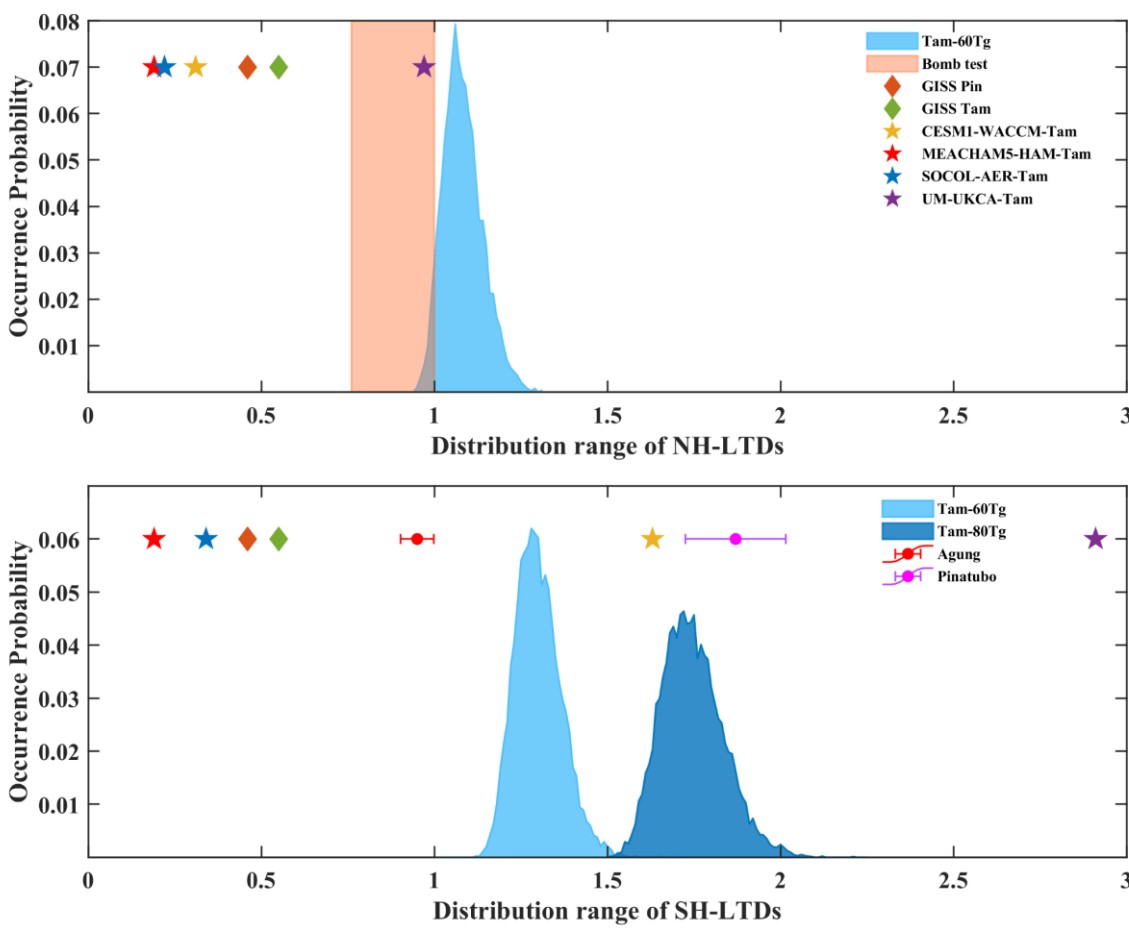


**Figure 5. Distribution of the conversion factors estimated from different methods.** The light and dark blue shadings represent the Monte Carlo simulated distribution of $LTD_T$ assuming a Tambora $SO_2$ injection of 60Tg and 80Tg, respectively. The LTDs estimated by different model simulations is represented by the diamonds and stars. The ice-core-based SH-LTD estimations for Agung and Pinatubo and their uncertainties are represented by the two circles.



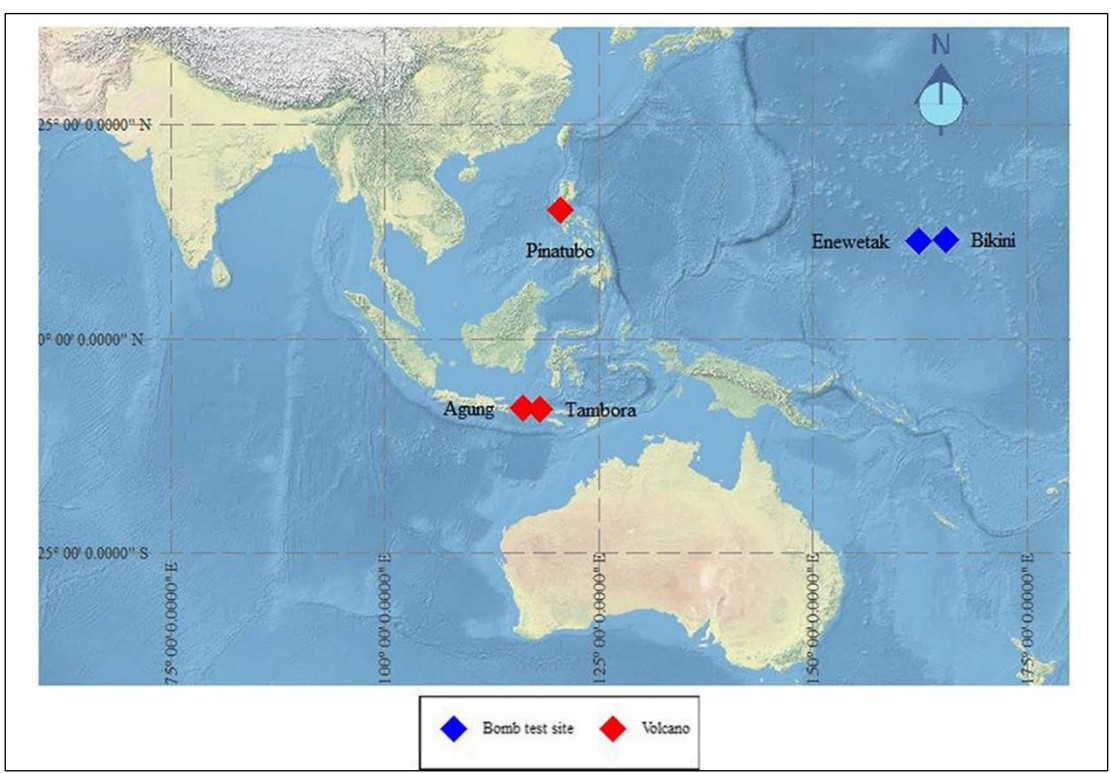


**Figure 6. Location of the Pinatubo, Agung, and Tambora volcano sites (red diamonds), and the 1952-54 US bomb test site sites of Enewatak and Bikini (blue diamonds).** The figure was drawn by plotting the site markers on a base map obtained from © Google Maps 2021.





**Table 1. Sulfate depositions for the 1815 Tambora eruption in the 25 Greenland ice cores** (data also available in the supplementary dataset DOI: 10.3974/geodb.2020.07.07.V1; Gao C C and Gao Y, 2020).

| Ice core sites | Latitude (°N) | Longitude (°W) | accumulation rate (m) | Tambora Dep. (kg km$^{-2}$) | Reference | Original source |
|---|---|---|---|---|---|---|
| Camp Century | 77.18 | 61.11 | 0.35 | 63 | Clausen & Hammer (1988) | / |
| North Central | 74.62 | 39.60 | 0.132 | 48 | Clausen & Hammer (1988) | / |
| Créte | 71.12 | 37.32 | 0.267 | 53 | Clausen & Hammer (1988) | / |
| Site A | 70.63 | 35.82 | 0.282 | 58 | Clausen & Hammer (1988) | / |
| Site B | 70.65 | 37.48 | 0.301 | 71 | Clausen & Hammer (1988) | / |
| Site D | 70.64 | 39.62 | 0.336 | 129 | Clausen & Hammer (1988) | / |
| Site E | 71.76 | 35.85 | 0.207 | 13 | Clausen & Hammer (1988) | / |
| Site G | 71.15 | 35.84 | 0.231 | 94 | Clausen & Hammer (1988) | / |
| Dye 3 | 65.18 | 43.83 | 0.5 | 54 | Clausen & Hammer (1988) | / |
| 4 B | 65.17 | 43.93 | N/A | 98 | Clausen & Hammer (1988) | / |
| 18 C | 65.03 | 44.39 | N/A | 25 | Clausen & Hammer (1988) | / |
| Dye 2 | 66.48 | 46.33 | 0.344 | N/A | Clausen & Hammer (1988) | / |
| Humboldt | 78.5 | 56.8 | 0.142 | N/A | Mosley-Thompson et al. (2003) | / |
| NASA-U | 73.8 | 49.5 | 0.333 | N/A | Mosley-Thompson et al. (2003) | / |
| D2 | 71.8 | 46.2 | 0.424 | 52 | Mosley-Thompson et al. (2003) | / |
| D3 | 69.8 | 44 | 0.488 | 85 | Mosley-Thompson et al. (2003) | / |
| Raven | 65.9 | 46.3 | 0.325 | 55 | Mosley-Thompson et al. (2003) | / |
| NGTb20 | 79 | 36.5 | 0.098 | 25 | Gao et al. (2007) | Bigler et al. (2002) |
| North GRIP | 75 | 43 | 0.152 | 37 | Gao et al. (2007) | / |
| GISP2 | 72.6 | 38.5 | 0.42 | 73 | Gao et al. (2007) | Zielinski, G. A. (1995) |
| Greenland site A | 70.8 | 36 | 0.267 | 27 | Gao et al. (2007) | Mosley-Thompson et al.(1993) |
| Greenland site T | 72.5 | 38.5 | 0.224 | 38 | Gao et al. (2007) | Mosley-Thompson et al.(1993) |
| 20D | 65 | 45 | 0.41 | 85 | Gao et al. (2007) | Mayewski et al, (1990) |
| Tunu2013 | 78 | 33.9 | 0.1 | 18 | Sigl et al. (2015) | / |
| NEEM2011S1 | 77.5 | 51.1 | 0.21 | 47.25 | Sigl et al. (2013) | / |








**Table 2. Sulfate depositions for the tropical eruptions of 1815 Tambora, 1963 Agung, and 1991 Pinatubo in the 33 Antarctic ice cores**

(data also available in the supplementary dataset DOI: 10.3974/geodb.2020.07.07.V1; Gao C C and Gao Y, 2020).


| Ice core sites | Latitude (° S) | Longitude (° E) | accumulation rate (m) | Tambora Dep. (kg km⁻²) | Agung Dep. (kg km⁻²) | Pinatubo Dep. (kg km⁻²) | Reference | Original source |
|---|---|---|---|---|---|---|---|---|
| Law Dome | 66.7 | 112.8 | 0.644 | 65 | 14 | 14 | Gao et al. (2007) | Palmer et al. (2002) |
| Dyer | 70.7 | 64.9 | 0.45 | 87 | 12 | N/A | Gao et al. (2007) | Cole-Dai et al. (1997) |
| G15 | 71 | 46 | 0.13 | 88 | N/A | N/A | Gao et al. (2007) | Moore et al. (1991) |
| Talos Dome | 72.8 | 159.1 | 0.081 | 38 | 3 | N/A | Gao et al. (2007) | Stenni et al. (2002) |
| Herc Neve | 73.1 | 165.5 | 0.119 | 19 | N/A | N/A | Gao et al. (2007) | Stenni et al. (2002) |
| Dome C | 75 | 123 | 0.031 | 8 | 8 | N/A | Gao et al. (2007) | E. Gautier et al.(2016) |
| DMLb32 | 75 | 0 | 0.061 | 28 | N/A | 12 | Gao et al. (2007) | Traufetter et al. (2004) |
| Siple Station | 76 | -84.3 | 0.55 | 118 | 29 | N/A | Gao et al. (2007) | Cole-Dai et al. (1997) |
| ITASE015 | 77.1 | -89.1 | 0.389 | 65 | 22 | 20 | Gao et al. (2007) | Dixon et al. (2004) |
| ITASE005 | 77.7 | -124 | 0.136 | 61 | 11 | 8 | Gao et al. (2007) | Dixon et al. (2004) |
| ITASE004 | 78.1 | -120.1 | 0.19 | 76 | 10 | 14 | Gao et al. (2007) | Dixon et al. (2004) |
| ITASE013 | 78.1 | -95.6 | 0.326 | N/A | 15 | 10 | Gao et al. (2007) | Dixon et al. (2004) |
| ITASE001 | 79.4 | -111.2 | 0.218 | 73 | 7 | 15 | Gao et al. (2007) | Dixon et al. (2004) |
| ITASE991 | 80.6 | -122.6 | 0.13 | 63 | 15 | 15 | Gao et al. (2007) | Dixon et al. (2004) |
| Plateau Rm | 84 | 43 | 0.04 | 20 | 8 | N/A | Gao et al. (2007) | Cole-Dai et al. (2000) |
| SP2001c1 | 89 | 0 | 0.08 | 61 | 8 | 3 | Gao et al. (2007) | Budner & Cole-Dai(2003) |
| SP95 | 89 | 0 | 0.078 | 60 | 10 | N/A | Gao et al. (2007) | Dixon et al. (2004) |
| PS1 | 89 | 0 | 0.08 | 68 | 0 | N/A | Gao et al. (2007) | Delmas et al. (1992) |
| PS14 | 89 | 0 | 0.08 | 61 | 8 | N/A | Gao et al. (2007) | Delmas et al. (1992) |
| DML05 | 75 | 0 | 0.062 | 28 | N/A | 11 | Sigl et al. (2015) | Traufetter et al. (2004) |
| B40 | 75 | 0 | 0.068 | 32 | 8 | 6 | Sigl et al. (2015) | / |
| DML07 | 75.6 | 3.4 | 0.059 | 41 | 7 | 3 | Sigl et al. (2015) | Traufetter et al. (2004) |
| NUS08-5 | 75.6 | 3.4 | 0.037 | 35 | 7 | 3 | Sigl et al. (2015) | / |
| NUS07-2 | 76.2 | 22.5 | 0.033 | 13 | N/A | N/A | Sigl et al. (2015) | / |
| DF01 | 77.4 | 39.7 | 0.027 | 29 | N/A | N/A | Sigl et al. (2015) | Motizuki et al. (2014) |
| DFS10 | 77.4 | 39.6 | 0.027 | 44 | 8 | 7 | Sigl et al. (2015) | / |
| Taylor Dome | 78 | 159 | 0.1 | 19 | N/A | N/A | Sigl et al. (2015) | Mayewski et al. (1996) |
| NUS07-5 | 78.7 | 35.6 | 0.024 | 28 | 4 | N/A | Sigl et al. (2015) | / |
| WDC06A | 79.5 | -112.1 | 0.21 | 78 | 13 | 15 | Sigl et al. (2013) | / |
| WDC05Q | 79.5 | -112.1 | 0.21 | 86 | 15 | 12 | Sigl et al. (2013) | / |
| DA2005 | 80.4 | 77.2 | 0.023 | 53 | N/A | N/A | Sigl et al. (2015) | Jiang, S et al (2012) |
| NUS07-7 | 82.1 | 54.9 | 0.03 | 9 | 19 | 3 | Sigl et al. (2015) | / |
| NUS08-4 | 82.8 | 19.8 | 0.036 | 17 | 12 | 5 | Sigl et al. (2015) | / |



**Table 3. Distribution statistics of the LTD factors of Tambora using Monte Carlo random sampling of selected number of ice core records**. The convergence rate is defined as the change in the range of μ±σ per increase of ice core number.

| Greenland; Tambora (1.075) | 8 (36%) 60Tg | 10 (45%) 60Tg | 12 (55%) 60Tg | 14 (64%) 60Tg | 16 (73%) 60Tg | 18 (82%) 60Tg | 20 (91%) 60Tg |
|---|---|---|---|---|---|---|---|
| Mean ($\times 10^9$ km$^2$) | 1.097 | 1.089 | 1.086 | 1.082 | 1.079 | 1.077 | 1.076 |
| Standard deviation | 0.166 | 0.135 | 0.111 | 0.091 | 0.075 | 0.056 | 0.038 |
| Convergence rate | NaN | 0.031 | 0.024 | 0.020 | 0.016 | 0.019 | 0.018 |
| Skewness[a] | 0.72 | 0.68 | 0.56 | 0.53 | 0.55 | 0.51 | 0.61 |
| Kurtosis[b] | 0.63 | 0.54 | 0.27 | 0.21 | 0.17 | 0.15 | 0.16 |
| Antarctica: Tambora (1.291) | 14 (44%) | 16 (50%) | 18 (56%) | 20 (63%) | 22 (69%) | 24 (75%) | 26 (81%) |
| Mean ($\times 10^9$ km$^2$) | 1.308 | 1.302 | 1.299 | 1.296 | 1.297 | 1.294 | 1.293 |
| Standard deviation | 0.138 | 0.118 | 0.101 | 0.089 | 0.077 | 0.066 | 0.054 |
| Convergence rate | NaN | 0.020 | 0.017 | 0.012 | 0.012 | 0.011 | 0.012 |
| Skewness | 0.97 | 0.65 | 0.47 | 0.49 | 0.38 | 0.46 | 0.41 |
| kurtosis | 3.49 | 1.49 | 0.55 | 0.72 | 0.13 | 0.22 | 0.02 |

[a] Skewness is defined as the asymmetry of the probability distribution of a random variable

[b] Kurtosis is defined as the steepness of the probability distribution of a random variable







**Table 4. Stratospheric volcanic aerosol loading to ice cap deposition conversion factors for tropical eruptions obtained in different studies** (data also available in the supplementary dataset DOI: 10.3974/geodb.2020.07.07.V1; Gao C C and Gao Y, 2020)**.**

| Method | For tropical eruptions based on Greenland ice cores ($\times 10^9$ km$^2$) | For tropical eruptions based on Antarctic ice cores ($\times 10^9$ km$^2$) | Reference |
|---|---|---|---|
| **Ice-core-based estimations** | | | |
| omb test calculation ($L_{\beta\text{-}1982}$) | 1.0-2.75 | | Clausen & Hammer (1988) |
| | 2.4 | | Zielinski (1995) |
| Bomb test calculation ($L_{\beta\text{-}2000}$) | 0.76-1.0 | | Gao et al. (2007) |
| Pinatubo observation ($L_P$) | | 1.0 | Gao et al. (2007) |
| | | 1.27-2.0[a] | Toohey & Sigl (2017) |
| Updated Pinatubo observation ($LTD_P$) | | 1.87±0.145 | This study |
| Agung observarion ($LTD_A$) | | 0.95±0.048 | This study |
| Tambora observation ($LTD_T$) | 1.08±0.056 | 1.29±0.066 | This study |
| [b]**Model simulations** | | | |
| GISS Pinatubo ($L_{GISS}$) | 0.46 | 0.46 | Gao et al. (2007) |
| GISS Tambora ($L_{GISS}$) | 0.55 | 0.55 | Gao et al. (2007) |
| CESM1-WACCM Tambora | 0.31 | 1.63 | Marshall et al (2018) |
| MEACHAM5-HAM Tambora | 0.19 | 0.19 | Toohey et al. ( 2013); Marshall et al. (2018) |
| SOCOL-AER Tambora | 0.22 | 0.34 | Marshall et al. (2018) |
| UM-UKCA Tambora | 0.97 | 2.91 | Marshall et al. (2018) |
| 4 model average ($BTD_T$) | 0.42 | 1.27 | Marshall et al. (2018) |

[a] The values are obtained by multiplying the original sulfate conversion factor of 1.2±0.3 km$^2$ by the factor 4/3, in order to scale the conversion factor from volcanic sulfate to sulfate aerosol.

[b] The four models used in Marshall et al. (2018) are coupled chemistry-climate models with chemical process and resolutions both superior to GISS model E, the obtained conversion factor is called Burden-to-deposition (BTD) factors.