# Peer review of "Uncertainties in the atmospheric loading to ice-sheet deposition for volcanic aerosols and implications for forcing reconstruction"

_Climate of the Past, 2021_

## Author Comment (AC1)

Reviewer #1:

(1) Many more cores could be included. The main additions in Antarctica are the Sigl et al. 2015 cores. Even the addition of ITASE cores on the two routes to South Pole from West Antarctica and Taylor Dome would provide valuable spatial information. It would also be interesting to assess multiple cores from the same location, such as South Pole or EDC, to better assess any single core uncertainty. It would be interesting for the authors to comment on whether improving the conversion factor is limited by simply the number of cores, or whether the location and resolution of analysis are also important.

*Response: We thank the reviewer for the insightful suggestion and questions.*

*We have tried to include as many cores as available, by periodically check the https://www.ncei.noaa.gov/products/paleoclimatology/ice-core data site. The dataset may appear to be incomplete because we choose not to use the data involving electrical conductivity measurements as they may introduce additional uncertainties. In this revision, we added two more Antarctic ice cores: WDC2020 that is recently available and EDC96 which was not included due to its approximate to Dome C. If there publicly available cores that are still missing in this analysis, please let us know.*

*We listed the six ITASE cores that have Tambora signals, if there are additional ITASE cores cover this period, we could love to know their names and how to obtain the data. We take and expand the reviewer's suggestion by examine the routes to South Pole from West Antarctica using the six ITASE cores and from the East Antarctic using five NUS cores. As shown in Table S1, there is no significant and consistent spatial pattern along both routes. We also compare the multiple cores within both South Pole and the coastal DML sites (Table S2) and the results suggest that volcanic signals in the South Pole cores show general agreement, while those in the DML cores may differ by a factor of 4 depending on the volcanic events. Another interesting pattern suggested by both tables is that signal magnitudes for Tambora show better agreement across the cores, likely because they are less influenced by the background noise.*

*Although additional ice core observations, especially from regions not covered in Figure 1&2 (if available), will certainly help to assess whether increasing the area-coverage of cores would improve the conversion factor, the latter is not limited by increasing the number of cores. Generally speaking, information about the nature of several representative eruptions, or the nature of volcanic cloud transport and aerosol*

*deposition pattern in the ice caps are more critical. Discussion of this has been added in the revision.*

(2) The analysis of why different sites have different magnitudes of sulfate could be developed more. In Antarctica, the authors find an average for East and West Antarctica and then do a simple area weighting of 80/20. There is much more that could be done to understand the spatial pattern and thus provide a more robust picture. In particular, sites with higher accumulation appear to have higher total sulfate. The authors sort of plot this in Figure 3, but there is almost no discussion of it. The authors should see how much of the variance in the total sulfate can be explained by accumulation. In East Antarctica, the sites are largely concentrated on the high plateau yet the coastal locations have a much larger total sulfate. This could introduce a significant bias.

*Response: The spatial pattern of the volcanic deposition in both Greenland and Antarctic, including the relationship between the accumulation rates and precipitation, has been examined and discussed heavily in our previous study (i.e., Gao, C., Oman, L., Robock, A., and Stenchikov, G. L. Atmospheric volcanic loading derived from bipolar ice cores accounting for the spatial distribution of volcanic deposition, J. Geophys. Res.-Atmos., 112, D09109, doi:10.1029/2006JD007461, 2007). That's why we did not spend much space discussing the spatial deposition pattern in this original submission. However, given the new ice core records added in this work and the reviewer's previous suggestion on examining the two routes to South Pole, we've added more detailed discussion on the spatial pattern of volcanic depositions.*

(3) The authors also need to address the robustness of aerosol loading estimates for Tambora. They indicate a range of 60-80 Tg, but seem to base their estimates off of the 60Tg number. It seems like using the 70 Tg mid-value would be more justifiable. This then translates to the uncertainties.

*Response: We choose the 60 Tg value mainly because it is the value used in the model simulations and the conversion factors could be compared. The uncertainty in Tambora aerosol loading does introduce another layer of uncertainty, but the uncertainty is linear as the conversion factor could be linearly scaled up to the 70 or 80 Tg.*

(4) Defining the different sources of uncertainty in the conversion factor, such as due to individual core representativeness, atmospheric loading uncertainty, and atmospheric/hemispheric transport (i.e. spatial variation), and volcano location. Then addressing each of these in a systematic way.

*Response:* *We sincerely thank the reviewer's insightful suggestion. The discussion on uncertainty in the conversion factor has been constructed into different sources as suggested in the revision.*

(5) Providing an example of how the new estimates change the volcanic forcing. Are the current estimates of volcanic forcing accurate? Does this new work increase or decrease the range in previously published volcanic forcing (e.g. Toohey and Sigl, 2017).

*Response:* *The new LTD is used to convert sulfate deposition to sulfate aerosol loading, when comparing with sulfate mass and volcanic forcing from eVolv2k (Toohey and Sigl, 2017), we need to multiply the results with a factor of 0.75. Therefore, taking the 1991 Pinatubo eruption for example, the new estimated SO2 injection is about 18.7Tg, within the range of SO2 release evaluated by Guo et al (2004), and volcanic forcing (-5.58W/m2) is approximately 86% of the previously that (-6.49 W/m2) in eVolv2k.*
*(Guo, S., Bluth, G. J. S., Rose, W. I., Watson, I. M., and Prata, A. J.: Re-evaluation of SO2 release of the 15 Jun 1991 Pinatubo eruption using ultraviolet and infrared satellite sensors, Geochem. Geophy. Geosy, 5(4), Q04001, doi.org/10.1029/2003GC000654,2004.)*

   *On the other hand, we would not consider the current estimates of volcanic forcing are accurate in general, and the new work implicates an increase in the range of previously published volcanic forcing, especially regarding IVI2 with contains no actual estimation of uncertainty range. The following comment has been added at the end of the conclusion:*
*"The results obtained from this study is a step forward to bring the conversion-induced uncertainty into the reconstruction framework, which hopefully also build a baseline for updating and improving the conversion of volcanic icecap-deposition to atmospheric-loading."*

---

## Author Comment (AC2)

**Reviewer #2:**

(1) L41 –The recommended forcing for PMIP4 is Toohey and Sigl (2017). Have these previous reconstructions been used in any CMIP6 simulations?

*Response: The Toohey and Sigl (2017) forcing reconstruction was based on Sigl et al. (2015). To be consistent with the PMIP4 references, we modify the text as copied below::*
*"the conversion factors have been utilized in the ice-core-based last millennium volcanic forcing reconstruction of Gao et al. (2008) and Toohey and Sigl (2017), which has been widely used in the CMIP5 and CMIP6 model simulations, respectively."*

(2) L50 –It would be useful to list what these 6 eruptions and the three methods are?

*Response: The information about the 9 Antarctic cores and the three methods have been added as copied below:*
*"Due to the limited number (0 in Greenland and 9 in Antarctic, including Law Dome, DML-B32, ITASE015, ITASE005, ITASE004, ITASE013, ITASE001, ITASE991, and SP2001c1) of ice core sulfate observations for the late 20th century, the conversion factor for Pinatubo is derived combining three methods (i.e., radioactive deposition from nuclear bomb tests, observation of Pinatubo sulfate deposition in eight Antarctic ice cores, and GISS Model E simulation of volcanic sulfate transport and deposition following the 1783 Laki, 1815 Tambora, 1912 Katmai, and 1991 Pinatubo eruptions, Gao et al. (2007))"*

(3) L125 –How different are these numbers compared to previous averages used to estimate the loadings?

*Response: The previous values of average deposition have been added and the revised text is copied below:*
*"The composites, after correcting for area-difference, show slightly smaller sulfate deposition in both Greenland (57 kg km$^{-2}$ in this study w.r.t. 59 kg km$^{-2}$ in Gao et al. (2007)) and Antarctic (47 kg km$^{-2}$ vs. 51 kg km$^{-2}$)."*

(4)  L155 - How realistic is it that the Monte Carlo characterization is the same for all low latitude eruptions?

*Response: The Monte Carlo characterization of the conversion factor for individual eruption depends on the number of available ice core observations and their recordings of the volcanic aerosols. We choose SH-LTD$_T$ and use its characterization to expand SH-LTD$_P$ and others, because it is obtained based on the largest collection of ice core records and the signals appeared to converge better than the other events. In another word, the possible range of SH-LTD$_P$ could be wider using Monte Carlo characterization of other events.*

*How realistic is the assumption can partially be evaluated by the general closeness between the volcanic sulfate aerosol distribution pattern and the precipitation pattern in both icecaps (Gao et al., 2007). It could also be subjected to change when the volcanic deposition is dominated by the dry deposition which maybe stochastic. However, we do not have such information to make the assessment. When ice core observations similar to number of Tambora observations become available for Pinatubo, Samala or other eruptions, we will be better equipped to evaluate the assumption. A brief discussion of how realistic is the assumption has been added in section 4.3.1.*

(5)  L174 – It would be useful to state the difference in the number of cores included for each ice sheet compared to Toohey and Sigl (2017)

*Response: The discussion has been revised by adding the difference in the number of cores among the different estimates, and as copied below:*

*"Toohey and Sigl (2017) also obtained a LTD estimation for Pinatubo using 4 Greenland and 18 Antarctic ice core records, and the result is 35% - 50% larger than LP. Our new ice core set includes most of the 22 records, accept SP04 whose Tambora signal is much smaller than the other four South Pole cores and the W10k record in Law Dome. And the newly obtained SH-LTD$_P$ is about 80% larger."*

(6)L184 – It is not clear exactly how the loading for Agung has been calculated and it

would be clearer to introduce the hemispheric partitioning at the beginning of this section.

*Response: The paragraph has been modified according to the reviewer's suggestion, as copied below:*

*"Agung volcano lies close to Tambora (Figure 6), while its 1963 AD eruption size is much smaller in terms of the sulfate aerosols (9.5Tg in Southern Hemisphere), therefore more sulfate aerosols may have stayed in the atmosphere longer and reached the ice sheets. By dividing the Southern Hemispheric loading with the Antarctic deposition of $10.25 kg/km^2$ averaged over the 24 ice core records available (Table 2), we obtain the SH-LTDA $= 0.95 \times 109 km^2$. According to Table 3, with 75 % precision the Monte Carlo simulation suggests the distribution range of SH-LTD$_A$ to be $(0.95 \pm 0.048) \times 109 km^2$."*

(6) L214, Table 4 and Figure 5 – Please check the units of the conversions - the model BTD factors in Marshall et al. (2018) are between deposition in kg $SO_4$ km$^{-2}$ and sulfate burden in Tg $SO_4$, not the aerosol loading. I think there is therefore a mismatch, and these should be scaled for comparison with the LTD factors.

*Response: We thank the reviewer for pointing out this unit issue. The BTD values have been scaled up in the revised Table 4 and Figure 5.*

(7) L246 – it would be useful to explicitly state why the sampling is important here and the implications for other eruptions

*Response: The following has been added:*

*"The Monte Carlo sampling procedure provides an estimation of LTDT uncertainty associated with the different number of ice core records used to calculate the conversion factor. Such information enables us to anticipate the level of uncertainty in LTD and subsequent forcing estimations for other eruptions with reduced number of ice core records."*

(8) Figure 5 – please add a and b labels. Why is Tambora 80 Tg not included on the top panel?

*Response: The labels have been added. The results for Tambora 80 Tg have been added to the top panel.*

---

## Author Comment (AC3)

**Reviewer #3:**

(1) L10: "CMIP5 and CMIP6 volcanic forcing" is ambiguous, since there were different forcings used in those 2 projects and within each project, for example for simulations of the historical era and on paleo timescales. More specificity is required.

*Response: The sentence has been changed to "This study revisits the Pinatubo-based LTD applied in the last Millennium volcanic forcing reconstruction that has been untilzed in the CMIP5 and CMIP6 simulations."*

(2) I do not believe the Monte Carlo procedure used herein constitutes a "model". A model is a mathematical framework that describes a system.

*Response: We agree and the reference to Monte Carlo has been changed to "Monte Carlo random sampling method/procedure".*

(3) L24: the second part of this sentence ("limitations in the conversion factor") is a tautology, needs more explanation.

*"This inverse reconstruction involves substantial uncertainties, due to the discrepancy in ice core volcanic deposition measurements and perhaps more importantly, the limitations in the conversion factor to transfer the ice core observation into the stratospheric volcanic sulfate loading."*

(4) L36: Reference needed for Pinatubo aerosol loading estimate

*Response: The two references for the original 15 Tg Southern Hemispheric sulfate aerosol loading estimate (listed below) have been added in the revision.*

*Bluth, G. J. S., C. C. Schnetzler, A. J. Krueger, and L. S. Walter (1993), The contribution of explosive volcanism to global atmospheric sulphur dioxide concentrations, Nature, 366, 327 - 329.*

*Krueger, A. J., L. S. Walter, P. K. Bhartia, C. C. Schnetzler, N. A. Krotkov, I. Sprod, and G. J. S. Bluth (1995), Volcanic sulfur dioxide measurements from the total ozone mapping spectrometer instruments, J. Geophys. Res., 100, 14057 - 14076.*

(5) L41: Again, need to be specific about the CMIP experiments which use this information.

*Response: The sentence has been changed to "the conversion factors have been utilized in the ice-core-based last millennium (850–1850CE) volcanic forcing reconstruction of Gao et al. (2008) and Toohey and Sigl (2017), which has been widely used in the CMIP5 and CMIP6 model simulations, respectively."*

(6) L52: "overlooked" is too strong, this is largely the motivation for the Toohey et al. (2013) and Marshall et al. (2018) studies.

*Response: The word has been changed to "occasionally studied by Gao et al. (2007) using mainly observation data, by Toohey et al. (2013) and Marshall et al. (2018) using model simulations".*

(7) L52: What is meant by "high depth resolution"?

*Response: We meant to say ice cores with "long term high resolution volcanic sulfate record". This ambiguity has been clarified in the revision.*

(8) L68: L cannot be either the mass of $SO_2$ or sulfate aerosols, of course you will get very different answers if you use one or the other (without some sort of conversion).

*Response: We thank the reviewer for pointing out the possible confusion. The unit has been limited to the one used in this study as copied "where L is the stratospheric volcanic mass loading (in* *Tg of sulfate aerosols**)"*

(9) L78: This sentence says that the authors have performed an extraction of the volcanic sulfate flux from these two ice cores independent of the work of Sigl et al. (2015). It would be helpful then to briefly present how the flux values the authors compute compare to those of Sigl et al. (2015).

*Response: Sigl et al. (2015) applied essentially the same methodology as Gao et al. (2007) therefore this study too, except for a stricter signal extraction threshold, i.e., 31yr trend + 3×MAD in Sigl et al. (2015) vs. 31yr trend + 2×MAD in this study. We*

*compared the magnitudes of the Tambora depositions in NEEM2011S1 and Tunu2013 using the two thresholds and found no difference. A brief comparison of the flux values has been added in the revision.*

(10) L86: Not quite clear if here again the authors have performed an independent estimation of the volcanic sulfate flux for these Antarctic cores? If so, a comparison with Sigl et al., (2015) would be quite useful.

*Response: Yes, we performed the same independent estimation for these Antarctic cores. This has been described more clear in the revision, and a brief comparison of the flux values with Sigl et al., (2015) has also been added as copied below:*

*"Sigl et al. (2015) applied essentially the same methodology as Gao et al. (2007) except for a stricter signal extraction threshold, i.e., three times the 31-year MAD instead of twice the MAD as Gao et al. (2007) and therefore also in this study. We compared the magnitudes of the Tambora, Pinatubo, and Agung volcanic depositions in these additional ice core records using the two thresholds and found almost no difference, except for NUS07-2 and NUS07-7 where the 3MAD threshold results in lightly smaller signals."*

(11) L91: How comparable the magnitudes of sulfur emission from Pinatubo and Agung are is arguable. In any case, does it really matter if they are, and does it matter that Agung is at a similar latitude as Tambora? If so, why?

*Response: In this section, we are trying to discuss the possible impacts of eruption magnitude and latitudinal locations on the conversion factor, with the very limited number of volcanic eruptions with observational information. When we say "they are the two recent events with comparable magnitudes" we are thinking about the order of magnitude as they are both smaller than Tambora (let alone Samala) but larger than the small eruptions of the later decades. Agung is at a similar latitude as Tambora but significantly smaller in magnitude, therefore may provide a case to detangle the influence of location vs. magnitude.*

*Nevertheless, in reality the locations and magnitudes of Tambora, Agung, and Pinatubo do not appear to matter much, given the other sources of influence (for example, the hemispheric partitioning) involved in the LTD. So we limit the referencing to these two events as "We also calculate the LTD factor of the 1963 Agung and 1991 Pinatubo eruptions, because they are the two recent* tropical *events with* moderate magnitudes and *some available observations."*

(12) L112: Where does the 60-80 Tg $SO_2$ range come from? Self et al. (2004) quotes 53-58 Tg $SO_2$, and Gertisser et al. (2012) don't seem to provide any independent estimate.

***Response:*** *The 60-80 Tg $SO_2$ range was actually borrowed from Marshall et al. (2018), because we are interested in compare our ice-core based results with those from the multi-model estimations. Thanks to the reviewer's question, we realize there is ambiguity in our description of the eruption and how we choose the size values. Therefore, in the revision we have modified the description as:*

*"The April 1815 eruption of Tambora (8.25° S, 118.00° E; Figure 5)* is one of the largest explosive eruptions in the Common Era (Self et al., 2004; Stoffel et al., 2015) and also *the most widely studied eruption in terms of ice core observation, model simulation, proxy reconstruction, and climatic and socioecological* aftermaths *(Luterbacher and Pfister, 2015; Raibie et al., 2016; Gao et al., 2017; Brönnimann et al., 2019)."*

*"Self et al. (2004) estimated a total of 53-58 Tg of $SO_2$ were released into the stratosphere by the Tambora eruption in 1815. Marshall et al. (2018) adopted the best estimate of 60 Tg $SO_2$ with a possible range of 30-80 Tg $SO_2$ in their multi-model simulations, after combining petrological, ice core, and aerosol process model estimations (Self et al., 2004; Gao et al., 2008; Stoffel et al., 2015). In order to better compare our results with the multimodel simulations, we take the best estimation of Tambora eruption size of 60 Tg $SO_2$ as the total amount of sulfate gases injected into the stratosphere and divide the values equally into each hemisphere."*

(13) L125: There's a logical problem here. If the flux to Antarctica and Greenland is

similar, to assume this means the aerosol partitioning is symmetric assumes a similar LTD factor for the two ice sheets. But then you use the assumption of even partitioning to calculate the LTD factor for the two ice sheets. This is circular.

*Response: The referencing to similar deposition flux in Antarctica and Greenland has been removed.*

(14) L145: The results stated here seem rather obvious results of the resampling procedure, but miss the point that the width of the distributions increases for smaller sample sizes, meaning that the uncertainty of a single sample (of some few ice cores) increases as n decreases. That the SD of the LTD decreases as 1/sqrt(n) simply confirms that the typical standard error of the mean (SD/sqrt(n)) is a suitable assumption for the data, but it is not clear if this has any physical meaning or utility for the present purposes.

*Response: Taking the reviewer's comments, we have emphasized the result that the width of the distributions increases for smaller sample sizes and how it relates to the uncertainty of the LTD estimation. The implication for the present purposes is also addressed by comparing the Monte Carlo sampling results for the number of cores available to representative periods of IVI2 or Sigl 2015 reconstructions and those for Tambora or Pinatubo deposition. Revisions addressing the two issues are copied below:*

*"The results show that, the distributions of LTDT with different ice-core sample sizes are approximately normal, and the width of the distributions increases for smaller sample sizes. This means that the uncertainty associated with a single sample increases as the number of ice core samples decreases."*

*Secondly, the standard deviation of LTDT decreases with 1/sqrt (N), confirming that the typical standard error of the mean (SD/sqrt(n)) is a suitable assumption for the data. In Figure 4 we also plot the standard deviation of the conversion factor if only the number of ice-core available to representative IVI2 or Sigl 2015 reconstructions were used solely, from which we can see significant reduction of the uncertainties as the number of ice core records increases to Pinatubo or Tambora level.*

(15) L150: To say that "the precision of LTD values is related to the limit in the number

of cores" is trivial, this is basic statistics. If you want to say "the precision of LTD values is ONLY related to the limit in the number of cores" then I would argue this is incorrect, because it depends on the random error of each ice core (from measurement noise or other factors), which is likely quite variable between different ice cores. If an ice core is particularly noisy, then adding it to the composite may increase the overall uncertainty. This analysis tells us nothing about the potentially very different errors of the individual ice cores.

*Response: The sentence has been removed. Instead, we focus the discussion on the changing nature of the standard deviation w.r.t the number of sampled ice cores. We also added in Figure 4 the Monte Carlo sampling results for the number of cores available to representative periods of IVI2 or Sigl 2015 reconstructions, so the discussion is more meaningful for the present purposes.*

(16) L151: The convention of quoting precision is unclear to me, I am used to percent precision for a value which is x+/-y as 100*y/x, so a smaller percent precision means more precise. That doesn't seem to be the case here.

*Response: The quoting precision here and those in Table 3 referring to the percentages of sampled number of ice core records with respect to the total available number of ice core records for Tambora. For example, 65% here is obtained by dividing 22 (the number of records sampled in this particular draw of Monte Carlo procedure) with 33 (total number of Antarctic ice core records). Therefore, a larger percent number means a higher number of available ice core records.*

*However, we feel from the reviewer's comment that this may induce confusion, and the sentence does not deliver much valuable information. Therefore, it is deleted in the revision. Explanation of these values in Table 3 has been added to the caption.*

(17) L182: I think your argument here has to do with aerosol particle size distribution, but this needs to be explained more clearly.

*Response: Yes, it is related to the aerosol particle size distribution among the three*

*eruptions. Specific explanation and related references have been added in the revision.*

(18) L193: "A series of…"?

*Response: Changed to "Series of nuclear bomb tests"*

(19) L216: If BTD is the same thing as LTD, please use the same name for it.

*Response: We prefer to keep BTD, because it can be easily differentiated from the ice-core based conversion factors and easily referenced back to its original source. If we change to LTD, then we have to add sub-notations to differentiate it from the ice-core based LTD, and we are afraid the sub-notations will be confused with the sub-notation of volcanic eruptions.*

(20) L267: This last statement makes no sense. If you use the LTD derived from Tambora on the ice core values for Tambora, you will of course get a loading estimate that is equal to the loading you used to calculate the LTD! You might as well just use the original loading estimate.

*Response: We agree with the comment and have revised the last statement as*
*"The results obtained from this study is a step forward to bring the conversion-induced uncertainty into the reconstruction framework, which hopefully also build a baseline for updating and improving the conversion of volcanic icecap-deposition to atmospheric-loading."*